# Did Government Expenditure on the Grain for Green Project Help the Forest Carbon Sequestration Increase in Yunnan, China?

**Ya'nan Lu [1,2], Shunbo Yao [1,2,*], Zhenmin Ding [1,2], Yuanjie Deng [1,2]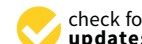 and Mengyang Hou [1,2]**

[1] College of Economics and Management, Northwest Agriculture and Forestry University, Xianyang 712100, China; luyanan@nwafu.edu.cn (Y.L.); huanglishanren@nwafu.edu.cn (Z.D.); dengyuanjie@nwafu.edu.cn (Y.D.); houmengyang@nwafu.edu.cn (M.H.)

[2] Economic and Environmental Management Research Center, Northwest Agriculture and Forestry University, Xianyang 712100, China

[*] Correspondence: yaoshunbo@nwafu.edu.cn

**Abstract:** Reasonably assessing the effectiveness of government expenditure on the Grain for Green project (GFG) in providing forest carbon sequestration would contribute to the development of China's forest carbon sequestration. Using the government expenditure data from the GFG in Yunnan Province from 2001 to 2015 and the MODIS Land Cover Type (MCD12Q1) time-series datasets, we calculated the forest carbon sequestration of various counties (cities or districts). The impacts of GFG government expenditure on forest carbon sequestration were empirically evaluated by the least squares dummy variables method (LSDV). The research results indicate that a 1% increase in government expenditure on the GFG yielded a 0.0364% increase in forest carbon sequestration. However, the effects of GFG government expenditure on forest carbon sequestration differed greatly in different areas because of the diversity of the natural environments, forest resource endowment, and government policies. If the initial forest endowment was not considered, the effectiveness of government expenditure on the GFG in providing forest carbon sequestration would have been overestimated. This study argues that, to improve the efficiency of GFG government expenditure in Yunnan Province, more investment should be made in regions with positive regression coefficients that have passed the significance *t*-test, such as Diqing Tibetan Autonomous Prefecture in the northwest, Baoshan City in the west, Honghe Hani and Yi Autonomous Prefecture in the south, and Wenshan Zhuang and Miao Autonomous Prefecture in the east.

**Keywords:** forest carbon sequestration; government expenditure; Grain for Green; least squares dummy variables method

## 1. Introduction

Climate change has become an important global environmental issue in recent years [1–3]. Increased concentrations of atmospheric greenhouse gas is the main cause of global warming [4,5]. As the largest land-based carbon pools [6], forest ecosystems reduce greenhouse gas emissions through carbon sequestration [7,8], which plays an important role in mitigating the effects of climate change [9,10]. The Kyoto Protocol states that afforestation and reforestation may be alternatives for reducing carbon emissions (IPCC, 2014) and the contribution of carbon sinks from woodland to total carbon sinks was over 90% in China from 1999–2014 [11]. Therefore, increasing forest area is undoubtedly an important measure to enhance terrestrial carbon sequestration [12] and reduce the concentration of greenhouse gas [13].

Since 1999, the Grain for Green project (GFG) was piloted in Shaanxi, Gansu, and Sichuan provinces, opening the prelude to the GFG in China. In 2002, the project was fully promoted. As the largest payments for ecosystem services in developing countries [14], the central and local governments in China mainly invested in special funds for the GFG in areas with severe ecological degradation through the way of fiscal transfer payment. The program attempted to increase the local forest and grass cover through afforestation in barren hills, closing hills for afforestation, and returning farmland to forest to improve the local ecological environment. According to the statistics, the national GFG government expenditure has reached 511.2 billion yuan, with project tasks covering 338,666.67 km$^2$ by the end of 2019[1]. Research assessing the effectiveness of payments for ecosystem services has been ongoing for many years. In terms of policy evaluation, scholars used forest cover [15], agricultural intensity [16], environmental services index [17], poverty alleviation index [18], and income and welfare indicators [19] as result variables to evaluate policy effects. The models for policy evaluation mainly included case analysis [20], OLS regression [21], randomized controlled trial [22], geographically weighted regression model [23], difference in difference [24], and matching method [25]. Because most ecosystem service payment programs were implemented according to administrative or political standards, there were systematic differences between project participants and non-participants [26], making it difficult to find a perfect control group when using matching method. However, it is difficult for other methods except matching method to separate the change of outcome variables caused by individual heterogeneity and other non-policy factors that are hard to observe, leading to certain errors in the estimated results of these studies. In terms of the effectiveness of forestry subsidies, the research results obtained by different methods at different spatial scales are significantly different. Studies on the effectiveness of forestry subsidies are rare. Some scholars employed DEA method to calculate and deconstruct forestry total factor productivity (TFP) growth in China; the study finds that the TFP shows some degree of negative growth and shows a negative contribution to the total output value of forestry, the added value of forestry [27]. Estimates based on social-cost and social-benefit analysis method indicate the annual average ecological cost-benefit ratio of the Grain for Green project is 38.11% in Nanjiang County of Sichuan Province [28]. Other estimates based on forest resource inventory data indicate that the increment of forest carbon storage will increase by 5~6% for each additional unit of forestry investment in fixed assets [29]. The above research has laid a foundation for the subsequent research in this field, but the existing research only focused on the total utilization efficiency of forestry investment funds without considering the individual differences in efficiency.

As the world's most extensive ecological restoration project with the largest amount of financial investment, the GFG resulted in increased forest and grass areas while providing a series of ecosystem services such as soil, water conservation, and microclimate regulation. Among these public services, providing carbon sequestration is one of few ecosystem services that has a recognized measurement method and can be traded via market transactions [30]. Therefore, this study chose forest carbon sequestration as the outcome variable to measure the policy effect to evaluate the environmental improvement effect of the policy. China's GFG has contributed approximately 25% of the biomass carbon sink in global carbon sequestration in 2000–2010 [12], which showed new hopes for us to make scientific evaluation on the effectiveness of the implementation of the project. Forest carbon sequestration research has been conducted for many years, and scholars have quantified the forest vegetation carbon stocks and carbon density at global [31,32], national [33,34], and regional scales [35,36]. The data used in these studies mainly comprised forest resource inventory data [37,38] and sample monitoring data [39]. The methods used included the continuous biomass expansion factor method [40], ecosystem modeling [41], average biomass method and volume biomass method [42]. Generally, gaining access to forest resource inventory data with spatial information is challenging, and the data are discontinuous

---

[1]　Forestry and Grassland Administration of the People's Republic of China, 2019. (http://www.forestry.gov.cn/main/225/20190905/163840772503997.html)



in time and cannot meet the needs of this study. Remote sensing satellite imagery of forest monitoring has greatly improved over the past few decades [43]. Hence, we selected remote sensing satellite images to monitor changes in vegetation cover in this study.

This study uses the example of Yunnan Province in China, which has a high forest coverage rate and a large forest carbon sequestration potential. Based on the MODIS (MCD12Q1) remote sensing satellite data, this study calculated the carbon sequestration by using Integrated Valuation of Ecosystem Services and Tradeoffs (InVEST) model to reflect the potential carbon sequestration contribution of project. Then, we chose the least squares dummy variables method (LSDV) to evaluate the effectiveness of providing forest carbon sequestration with the GFG government expenditure. Remote sensing data in successive years were used to calculate carbon sequestration to avoid discontinuities in time when using forest inventory data. This study mainly answered the following questions: 1. What was the relationship between forest carbon sequestration and financial expenditure of the GFG, that is, the change of forest carbon sequestration resulting from each additional unit of capital investment in the study area? 2. Were there significant regional differences in the increase of forest carbon sequestration based on GFG government expenditure? 3. How can GFG government expenditure be reasonably adjusted for various regions when the goal is to increase forest carbon sequestration? The answers will contribute to the development of China's forest carbon sequestration and provide important decision-making background for the new round of the GFG.

## 2. Data Source and Model Establishment

### 2.1. Overview of the Study Area

Yunnan is located between east longitude 97°31'39"~106°11'47" and north latitude 21°08'32"~29°15'08" (see Figure 1). It contains the upper and middle reaches and main catchment areas of the Nu, Jinsha, Lancang, Irrawaddy, and other rivers, and its ecological status is very important. The Yunnan forest coverage rate reached 59.7% by the end of 2018, 2.76 times the national average (21.63%) and ranked 7th in China (People's Daily Online, 2018)[2]. Among the 129 counties (cities or districts) in the province, mountainous areas account for up to 99% of 18 counties, and the proportion of mountainous areas in other regions is more than 70%, with the exceptions of Wuhua and Panlong [44]. Soil erosion is among the most serious ecological environment problems in the region, seriously restricting the sustainable development of local societies and economies as well as the middle and lower reaches of the Yangtze River and the Pearl River. In order to alleviate ecological problems, a pilot program of the Grain for Green project was started in 2000 to alleviate ecological problems and was extended to the entire province in 2002. From the beginning of the program to 2015, Yunnan spent 12.14 billion yuan on the GFG and has returned 12,092.87 $km^2$, including 3554 $km^2$ of returned farmland for afforestation, 7068.87 $km^2$ of barren mountains and wasteland for forestry, and 1470 $km^2$ of closed mountains for afforestation (Returning Farmland to Forest Office of Yunnan Province), accounting for 29.39%, 58.45%, and 12.16%, respectively, of the GFG area.

---

[2]  People's Daily Online, 2018. (http://yn.people.com.cn/n2/2018/1211/c378439-32393113.html)

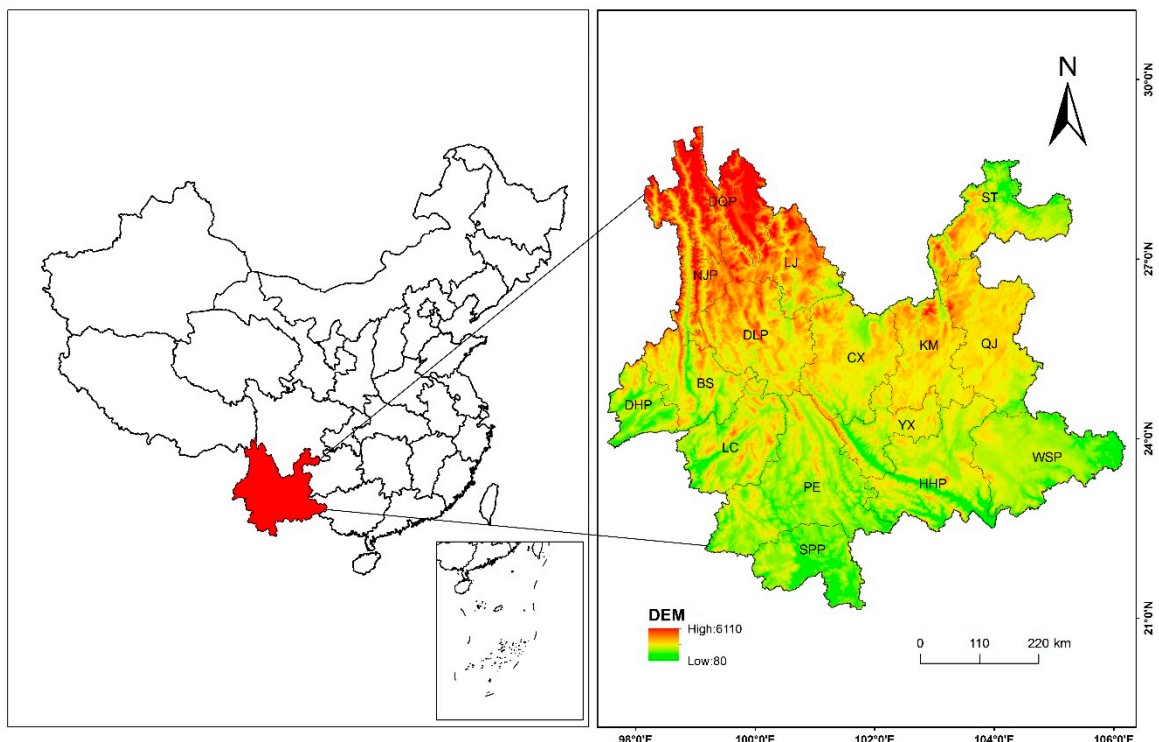

**Figure 1.** The location of Yunnan province (KM: Kunming City; QJ: Qujing City; YX: Yuxi City; BS: Baoshan City; ST: Shaotong City; LJ: Lijiang City; PE: Puer City; LC: Lincang City; CX: Chuxiong Yi Autonomous Prefecture; HHP: Hani-Yi Autonomous Prefecture of Honghe; WSP: Wenshan Zhuang and Miao Autonomous Prefecture; SPP: Dai Autonomous Prefecture of Sipsongpanna; DLP: Dali Bai Autonomous Prefecture; DHP: Dehong Autonomous Prefecture; NJP: Nujiang of the Lisu Autonomous Prefecture; DQP: Diqing Tibetan Autonomous Prefecture).

*2.2. Model Building*

This study used the least squares dummy variables method (LSDV) [45,46] to investigate the effect of government expenditure on forest carbon sequestration. The model had some advantages: First, the model can observe variables that do not change with time. Second, this model can be used to obtain clustering robust standard errors and solve the model estimation problems caused by heteroscedasticity and missing variables. The omission of variables is a common cause of bias in empirical results. The least squares dummy variables method can overcome this problem by introducing the individual dummy variables. Third, the model can examine the individual heterogeneity of multiple counties (cities or districts) by introducing the interaction term between individual dummy variables and key explanatory variables. Since the main purpose of this study is to effectively uncover the marginal contribution of the GFG government expenditure on forest carbon sequestration, other factors such as annual temperature, average annual precipitation, slope, aspect, per capita GDP, and land urbanization rate were selected as control variables. In the first place, to show the average impact of the GFG government expenditure on forest carbon sequestration, the study constructed a regression model as follows:

$$ln\ carbon_i = \alpha_0 + \beta\ ln\ capital_i + \sum \gamma_j Z_{ij} + \sum id + \varepsilon_i \tag{1}$$

Secondly, to explore regional differences in the effect of GFG implementation, the study introduces the interaction between individual dummy variables and the key explanatory variable of government expenditure and constructed a model as follows:

$$ln\ carbon_i = \alpha_0 + \sum \gamma_j Z_{ij} + \sum \beta_i id\#\ ln\ capital + \sum id + \varepsilon_i \tag{2}$$

Finally, the increase of the forest carbon sequestration may be self-dependent, i.e., vegetation growth is largely affected by initial forest resource endowment (Initial forest resource endowment refers to forest carbon sequestration in the previous year). To fully consider the "early dependence characteristics" of explained variables, this study introduces the first-order lag term of explained variables and constructed a panel regression model as follows:

$$ln\ carbon_i = \alpha_0 + \alpha_1 L\ ln\ carbon_i + \sum \gamma_j Z_{ij} + \sum \beta_i id\#\ ln\ capital + \sum id + \varepsilon_i \qquad (3)$$

The subscript *i* represents each county (city or district) of Yunnan. $lncarbon_i$ represents the amount of forest carbon sequestration, which is the explained variable; $lncapital_i$ is the accumulated input of the GFG government expenditure, which represents the core explanatory variable; $Llncarbon_i$ represents the first order lag term of forest carbon sequestration. $Z_{ij}$ represents a conventional explanatory variable that may affect forest carbon sequestration, also known as a control variable. Where $Z_{i1}$ represents annual temperature of county *i*, $Z_{i2}$ represents average annual precipitation of county *i*, $Z_{i3}$ represents *slope* of county *I*, $Z_{i4}$ represents *aspect* of county *i*, $Z_{i5}$ represents per capita GDP of county *i*, $Z_{i6}$ represents land urbanization rate of county *i*. *id#lncapital* represents the product of classified variables (id, the value range is 1 to 129) and continuous variables (*lncapital*). $\alpha$, $\gamma$, and $\beta$ are the estimation coefficients of the corresponding variables. $\sum id$ represents the individual dummy variables. $\varepsilon_i$ is a random disturbance term.

### 2.3. Variable Selection and Description

#### 2.3.1. Estimation of Forest Carbon Sequestration

This study chose the widely used InVEST model [47–49] to calculate the carbon sequestration. The model mainly calculated the carbon storage of different vegetation types at different periods according to various vegetation cover data and the carbon density coefficients of the five corresponding carbon pools. The vegetation cover data are from a MODIS Land Cover Type Product (MCD12Q1) dataset from 2001 to 2015 provided by NASA. The data have 500 m spatial resolution and include 5 legacy classification schemes. These schemes include the 17-class International Geosphere-Biosphere Programme classification (IGBP); the 14-class University of Maryland classification (UMD); a 10-class system used by the MODIS LAI/FPAR algorithm (LAI); an 8-biome classification (BGC); and a 12-class plant functional type classification (PFT) [50]. The IGBP land cover classification system reflects the physiological parameters of the surface in the land cover as well as the importance of vegetation status in the land cover and can meet the needs of this study. We used NASA's HDF-EOS to GeoTIFF (HEG) downloadable conversion tool to transform and work with MODIS Land Cover Type Product (MCD12Q1) data. This dataset accuracy, obtained by classification with the IGBP classification method, was 74.8%, of which 72.3–77.4% reached a 95% confidence level [50].

The five major carbon pools included underground biomass, aboveground biomass, dead organic matter, soil, and carbon pools of wood products or related wood products [51]. On the one hand, Yunnan Province had issued a ban on logging before 2000, so the amount of timber cut was negligible. On the other hand, China's current timber market is incomplete, and data such as the timber attenuation rate are difficult to obtain. Therefore, only the first four carbon pools were considered in this study. According to the research needs, this study only considered the carbon storage of the forest. The carbon storage calculation principle of the InVEST model is as follows:

$$C = C_{above} + C_{below} + C_{dead} + C_{soil} \qquad (4)$$

As shown, *C* represents the total carbon storage, $C_{above}$ represents aboveground carbon storage, $C_{below}$ represents underground carbon storage, $C_{dead}$ represents the dead organic matter carbon storage, and $C_{soil}$ represents soil carbon storage. Each carbon storage type is a product of forest area and the corresponding carbon density. Vegetation types required in this study include coniferous forest,

evergreen broadleaf forest, deciduous broadleaf forest, mixed forest, and shrubland. The carbon density coefficient corresponding to the vegetation cover is based on previous research results [52], adjusted slightly. Finally, Table 1 shows the vegetation cover types and corresponding carbon density coefficients.

**Table 1.** Carbon density parameter table in Yunnan (unit: t/ha).

| Vegetation Coverage Type | Land Cover Type | C_above | C_below | C_dead | C_soil |
|---|---|---|---|---|---|
| Coniferous Forest | Evergreen Needleleaf Forests Deciduous Needleleaf Forests | 51.87 | 15.03 | 4.10 | 21.84 |
| Evergreen Broadleaf Forest | Evergreen Broadleaf Forests | 36.85 | 7.37 | 2.80 | 32.30 |
| Deciduous Broadleaf Forest | Deciduous Broadleaf Forests | 48.85 | 9.77 | 1.90 | 11.7 |
| Mixed Forest | Mixed Forests | 30.63 | 6.91 | 1.80 | 22.91 |
| Shrubland | Closed Shrublands Open Shrublands Woody Savannas | 6.85 | 7.60 | 2.57 | 12.88 |

In this study, land cover data and carbon density data, which are the necessary data of the InVEST model, were selected to calculate the forest carbon sequestration in Yunnan from 2001 to 2015. In order to simplify the description, this study briefly introduces its operation steps with 2001 as an example, and the calculation method is the same for other years. Firstly, the land cover data processed by HDF-EOS to Geotiff (HEG) conversion tool were reclassified into six categories, namely coniferous forest, evergreen broadleaf forest, deciduous broadleaf forest, mixed forest, shrubland, and other land types. Because this study only studies forestry carbon sequestration, for the sake of easy operation, the carbon density of four carbon pools in other land types was set to 1. Secondly, the reclassified land cover data and carbon density data were input into the InVEST model to obtain the distribution map of forest carbon sequestration in Yunnan Province. Lastly, the carbon sequestration of each land cover type in 129 counties (cities or districts) in Yunnan was calculated by the regional statistical function of ArcGIS. Taking Kunming as an example, the forest carbon sequestration in 2001 was the sum of coniferous forest, evergreen broadleaf forest, deciduous broadleaf forest, mixed forest, and shrubland. Through the above methods, the forest carbon sequestration of 129 counties (cities or districts) in Yunnan from 2001 to 2015 was obtained.

2.3.2. Government Expenditure on the GFG

The GFG government expenditure data in Yunnan from 2000 to 2015 were derived from the statistical data of the GFG Office of the Forestry Department of Yunnan Province. Because the forest carbon sequestration in this study was cumulative, the actual capital stock of each county (city or district) in that year was used as the proxy variable for the government expenditure of the GFG. Based on previous research results [53], this study used the perpetual inventory method to calculate the actual capital stock using the following calculation formula:

$$capital_{it} = I_{it} + capital_{i,t-1}(1 - \delta) \tag{5}$$

In the formula, $capital_{it}$ is the capital input stock on the GFG of the county (city or district) $i$ in the year $t$. $I_{it}$ is the annual GFG capital input of the county (city or district) $i$ in the year $t$. $Capital_{i,t-1}$ is the capital input stock on the GFG of the county (city or district) $i$ in the year $t-1$. $\delta$ is the economic depreciation rate. When calculating the actual capital stock, 2001 was used as the base period, and the economic depreciation rate was determined to be 7.5% [53]. Additionally, to eliminate heteroscedasticity in

the sequence of government expenditure on the GFG and forest carbon sequestration, the two are logarithmically expressed by *lncapital* and *lncarbon*, respectively. The more the GFG government expenditure, the larger the forest area, which is more conducive to the increase of forest carbon sequestration. Thus, the sign (+/−) of coefficient of the expected GFG government expenditure is positive (+).

### 2.3.3. Control Variables

To improve the accuracy of the model estimation results, this study introduced relevant variables as control variables considering natural and socio-economic aspects. Studies have shown that natural environmental factors such as precipitation and temperature are indispensable to related research on forest carbon sinks [54]. The annual average temperature (*Tem*) and annual precipitation (*Pre*) data used in this study came from National Meteorological Information Center (http://data.cma.cn). According to the annual data on temperature and precipitation and the longitude, latitude, and altitude information of each meteorological station, the inverse distance weight spatial interpolation method was used in ArcGIS10.3 to obtain a spatially continuous meteorological grid with a consistent pixel size for the study area. Finally, the annual average temperature and annual precipitation data for each county (city or district) in the study area were obtained through Zoning Statistics in ArcGIS 10.3; both were expected to have positive (+) sign of the coefficients. Slope (*Mslp*) and aspect (*Asp*) data were extracted from 250 m elevation (digital elevation model, DEM) data from the Resource and Environmental Data Cloud Platform (http://www.resdc.cn). The greater the slope, the less human activity, which is more conducive to vegetation growth. In this study, the greater the treated slope aspect value, the closer it was to a sunny slope; thus, the sign (+/−) of coefficient of the expected slope and aspect are both positive (+).

The socio-economic factors per capita GDP (*RGDP*) and land urbanization rate (*Pstr*) were introduced. Rapid development of the economy meant that cropland, forest, and grassland were used for urban and rural construction land, resulting in decreased in forest area and carbon sinks [55]. Therefore, the signs (+/−) of the coefficients of per capita GDP and land urbanization rate were expected to be negative (−). Per capita GDP is a non-stationary variable [56–58]; therefore, the per capita GDP growth rate (*RGDP*) was calculated instead of per capita GDP to reflect the economic development level of each region. The per capita GDP data came from the Yunnan Statistical Yearbook. Considering the impact of inflation, this study used 2001 as the base period and revised the per capita GDP using the Yunnan consumer price index. The land urbanization rate was expressed as the ratio of urban and built-up lands to the total area of the county. The urban and built-up land area and the total area of the county (city or district) were extracted from the MODIS Land Cover Type Product (MCD12Q1) dataset for the period from 2001 to 2015. This dataset was processed firstly through NASA's HDF-EOS to GeoTIFF (HEG) downloadable conversion tool. The tool of Zoning Statistics was then used in ArcGIS10.3 to obtain the urban and built-up land area and total area of the county (city or district). The description and statistical results of the main variables are shown in Table 2.

**Table 2.** Statistical description of main variables.

| Variable | Symbol | Variable Description/Unit | Mean | Standard Deviation |
|---|---|---|---|---|
| Forest carbon sequestration | carbon | Calculation and Extraction/Pg C | 8.4535 | 9.1551 |
| Government expenditure on the GFG | capital | Yunnan Provincial Forestry Department/RMB 10,000 Yuan | 4323.1440 | 3289.0280 |
| Annual precipitation | Pre | Interpolation and Extraction/mm | 101.1543 | 25.5772 |
| Average annual temperature | Tem | Interpolation and Extraction/°C | 16.9494 | 2.2586 |
| Slope | Mslp | Data Extraction/° | 6.7381 | 2.7284 |
| Aspect | Asp | Data Extraction/° | −10.4058 | 8.1490 |
| The growth rate of per capita GDP | RGDP | (per capita $GDP_t$ − per capital $GDP_{t-1}$)/ per capita $GDP_{t-1}$/100% | 0.1247 | 0.2655 |
| Land urbanization rate | Pstr | Urban and built-up land area/county total area/100% | 0.0190 | 0.0432 |

## 3. Results

### 3.1. Changes in Government Expenditure

By 2015, the total accumulated investment on the GFG in Yunnan was about 12.138 billion yuan and included investment in capital construction, special subsidies from the state finance, and provincial financial subsidies. Capital construction investment included seed base construction, nursery construction, seedling infrastructure construction, seedling afforestation subsidies, and preliminary work funds. From 2000 to 2015, the annual total investment on the GFG of 129 counties (cities or districts) in Yunnan showed an increasing trend before levelling off and finally decreasing. The project had 123 million yuan total investment in 2000 and reached a peak value of 1.442 billion yuan in 2005, declining yearly after that (see Figure 2). The year-by-year investment curve shows that investment in the GFG was concentrated in the years from 2003 to 2009 and has dropped sharply since 2010. The stock of GFG investment first showed a rising trend from 2000 to 2010 and reached a peak value of 8.533 billion yuan in 2010, then declined yearly after that.

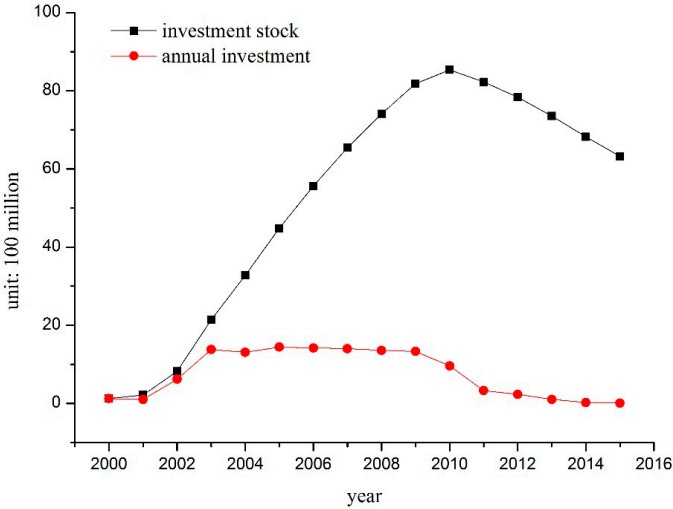

**Figure 2.** Changes in government expenditure on the Grain for Green project (GFG) from 2000 to 2015.

### 3.2. Changes in Forest Carbon Sequestration

Figure 3 shows the changes in forest carbon sequestration in Yunnan. From 2001 to 2015, the total forest carbon sequestration showed an upward trend, increasing from 1.07 Pg in 2001 to 1.15 Pg in 2015 at an average annual growth rate of 0.52%. The change trend of forest carbon sequestration showed strong inertia. For example, in both 2001 and 2015, the counties (cities or districts) with the top three largest forest carbon sequestration were Shangri-La City, Mengla County, and Jinggu Yi and Dai Autonomous County, from highest to lowest (see Figure 3a,b). The three counties with the largest increases in forest carbon sequestration during the study period were Lancang Lahu Autonomous County, Ninglang Yi Autonomous County, and Longyang District. Yanshan County, Luxi county, and Huize County had the highest (exceeding 90%) forest carbon sequestration growth rates of 261.61%, 97.24%, and 93.94%, respectively. Of 30 counties (cities or districts) that showed decreased forest carbon sequestration, Menghai County ($-1.57 \times 10^{-3}$ Pg), Mangshi ($-6.27 \times 10^{-4}$ Pg), and Simao District ($-6.22 \times 10^{-4}$ Pg) had the greatest decreases. The three counties with the highest reduction rate were Kaiyuan City (19.76%), Guandu District (11.29%), and Xishan District (7.99%). Generally, the counties (cities or districts) with the highest forest carbon sequestration were distributed in the Northwest and Southwest Yunnan Province, while the counties (cities or districts) with the highest forest carbon sequestration growth rates were distributed in Eastern Yunnan Province.

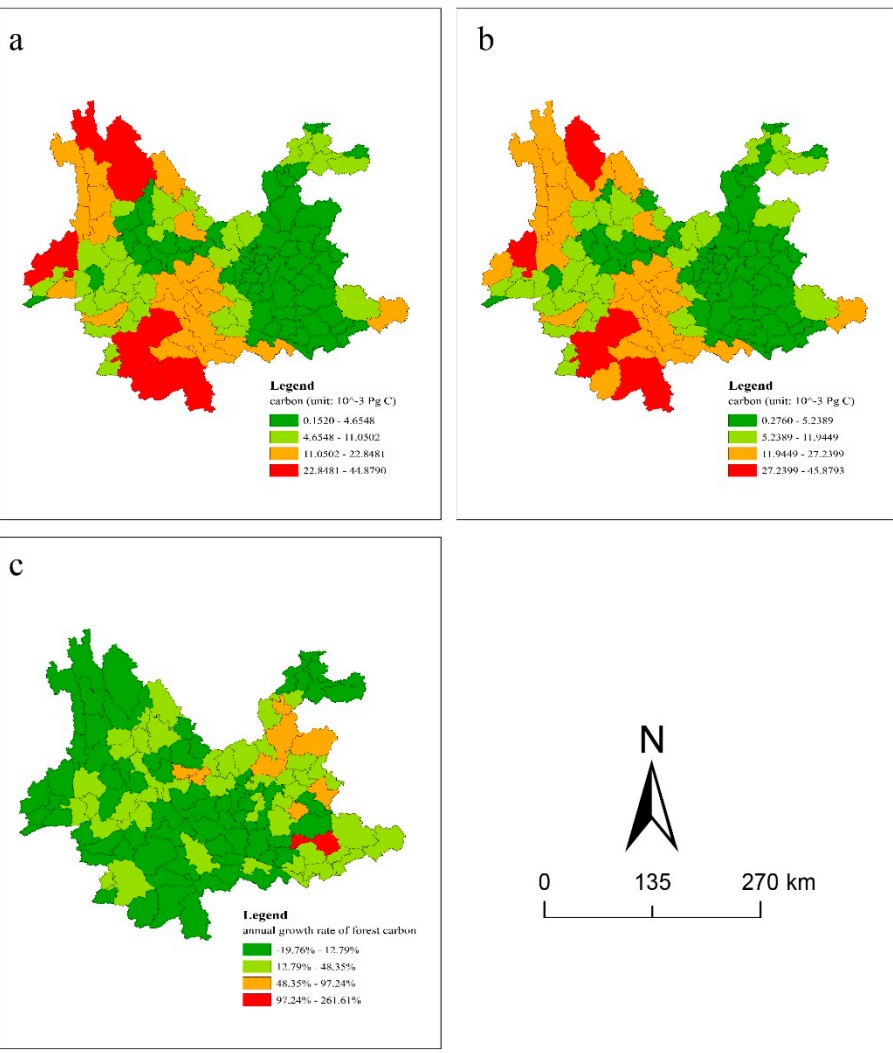

**Figure 3.** Distribution of forest carbon sequestration in Yunnan; (**a**) 2001; (**b**) 2015; (**c**) annual growth rate of forest carbon sequestration from 2001 to 2015.

### 3.3. Model Estimation Results

　　Using panel data from 129 counties (cities or districts) in Yunnan from 2001 to 2015, the regression Model (1) was used firstly to measure the average effect of the GFG in Yunnan. The larger the value, the better the effect of the GFG. Model (1) of Table 3 shows the average impact of government expenditure on forest carbon sequestration. The key explanatory variable GFG government expenditure has a positive impact on forest carbon sequestration with a significance level of 1% and an elasticity coefficient of 0.0364. In other words, a 1% increase in investment led to a 0.0364% increase in forest carbon sequestration, indicating that the GFG effectively increased the forest carbon sequestration and improved the regional ecological environment. However, considering the large differences in natural environments and social and economic conditions, the effect of the GFG policy is inevitably different among regions in Yunnan. As shown in Figure 3c, the annual growth rate of forest carbon sequestration in each county (city or district) over the past 15 years is greatly different. The study introduces the interaction between individual dummy variables and the key explanatory variable of government expenditure to explore regional differences in the effect of GFG implementation. The regression results are shown in Model (2) of Table 3 and Figure 4a. The estimation results showed that the coefficients of most individual dummy variables and interactions between individual dummy variables and key explanatory variables (*lncapital*) were significant, and the signs (+/−) of the regression coefficients conformed to both theory and expectation, indicating that the variable coefficient model was set appropriately [59] (see supporting information Appendix A for more details).

**Table 3.** Test on the impact of GFG government expenditure on forest carbon sequestration.

| Variable | Model (1) | Model (2) | Model (3) |
|---|---|---|---|
| Llncarbon | — | — | 0.9040 *** (61.69) |
| lncapital | 0.0364 *** (9.56) | — | — |
| Pre | 0.0011 *** (7.37) | 0.0012 *** (9.31) | 0.0005 *** (11.58) |
| Tem | 0.0265 *** (5.34) | 0.0279 *** (7.21) | 0.0060 *** (4.11) |
| Mslp | 0.4937 *** (29.98) | 0.5442 *** (20.95) | 0.0520 *** (3.73) |
| Asp | 0.0843 *** (3.14) | 0.0600 (1.51) | −0.0049 (−0.28) |
| RGDP | −0.0042 (−0.45) | −0.0027 (−0.37) | 0.0001 (0.05) |
| Pstr | 3.6724 *** (2.72) | 3.7750 *** (2.92) | 0.4786 (1.59) |
| Control of the individual | YES | YES | YES |
| Regional difference in efficiency | — | Figure 4a | Figure 4b |
| _cons | 12.5972 *** (222.09) | 12.4858 *** (58.30) | 1.0380 *** (4.83) |
| Observations | 1806 | 1806 | 1806 |
| R-squared | 0.9963 | 0.9981 | 0.9998 † |

† *, *** represents 10%, 5%, 1% of the significance level; the value in brackets of estimated coefficient is its *t*-value.

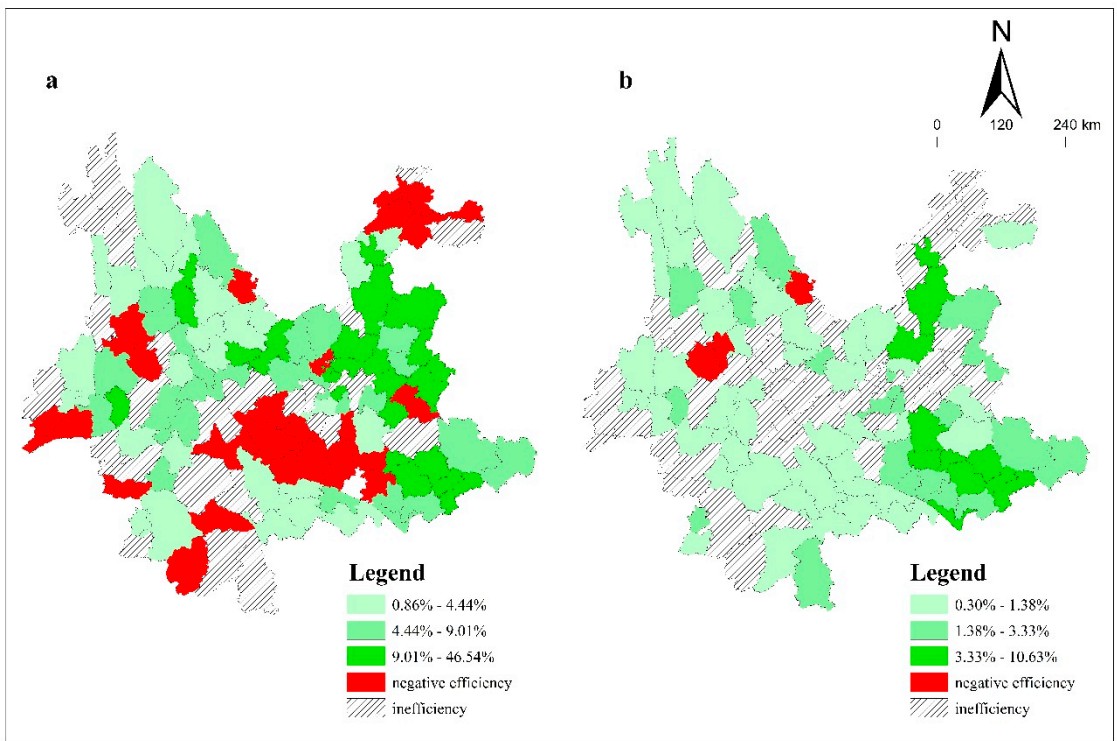

**Figure 4.** Distribution of government expenditure efficiency on the GFG. Negative efficiency represents that the regression coefficients are negative (−) and pass the significance *t*-test; inefficiency represents that the regression coefficients do not pass the significance *t*-testing. (**a**) partial regression results for Model (2); (**b**) partial regression results for Model (3).

As Figure 4a shows, the government expenditure of the GFG affected forest carbon sequestration differently in different regions. In the north and southeast parts of Yunnan, 75 counties (cities or districts) represented by Yanshan County, Huize County, and Heqing County had positive efficiencies (the sign of the regression coefficients is positive (+), and it passed the significance *t*-testing), indicating that GFG implementation was conducive to the increase of forest carbon sequestration in those regions. However, 27 counties (cities or districts) including Dongshan and Hongta District were inefficient (the regression coefficients do not pass the significance *t*-testing). These inefficient counties (cities or districts) accounted for 20.93% of the total and had scattered distribution. This shows that government expenditure on the GFG did not play a significant role in promoting forest carbon sequestration in these regions. A total of 27 counties (cities or districts) including Daguan County, Kaiyuan City, and Yunlong County had negative efficiency (the sign of the regression coefficients is negative (−), and they passed the significance *t*-testing); most of these counties (cities or districts) were distributed north of Zhaotong City, south of Chuxiong Yi Autonomous Prefecture, the western part of Yuxi City, the northern part of Honghe Hani and Yi Autonomous Prefecture, the southwest of Dali Bai Autonomous Prefecture, and the southwest of Dehong Yi and Jingpo Autonomous Prefecture. Forest carbon sequestration decreased rather than increased in these areas as capital investment in the GFG increased, which appears to be contrary to the laws of natural development.

Considering the particularity of the forest growth, the increase of the forest carbon sequestration may be self-dependent, i.e., vegetation growth is largely affected by initial forest resource endowment, which may explain the differences in efficiency. Furthermore, in the project area of GFG, vegetation restoration should result from the combined effects of ecological policies and local forest resource endowment. Initial forest endowment was not considered in Model (2), and important variables may be missing. Therefore, to investigate the effects of policy implementation, we introduced the first-order lagged variable of forest carbon sequestration to the model to remove the influence

of the initial forest endowment on forest carbon sequestration. The regression results are shown in Model (3) of Table 3 and Figure 4b. By controlling the initial forest resource endowment, the regression coefficient of the first-order lagged variable of forest carbon sequestration is positive (+) at a 1% significance level, which indicates that the increase of the forest carbon sequestration does have path dependence. Past forest resource endowment positively affects the implementation effects of current policies; with the same input, the better the initial forest resource endowment (initial forest endowment refers to forest carbon sequestration in the previous year), the higher the efficiency of the capital use and the greater the increase of forest carbon sequestration. However, there was a decline in overall regression coefficient level, indicating that if the initial forest resource endowment was not considered, the promotion effect of the forest carbon sequestration by GFG capital investment would be overestimated. Further analysis of the regression results shows that a total of 75 counties (cities or districts) had positive (+) estimation coefficients and passed significance *t*-testing. Those counties (cities or districts) were mainly distributed in Diqing Tibetan Autonomous Prefecture in the northwest, Baoshan City in the west, Honghe Hani and Yi Autonomous Prefecture in the south, and Wenshan Zhuang and Miao Autonomous Prefecture in the east. The government expenditure of the 51 counties (cities or districts) had no significant positive impact on forest carbon sequestration. Those counties (cities or districts) showed a zonal distribution, from Zhaotong City in the northeast, Dehong Dai and Jingpo Autonomous Prefecture, Lincang City and Pu'er City in the southwest, and the entire central region.

This study mainly used Model (3) to analyze the size and direction of control variables affecting forest carbon sequestration. The estimation coefficient of annual precipitation was positive (+) and significant, but the coefficient was very small. This result is consistent with a previous study reported by Dominik and William [60], who identified a weak positive relationship between carbon sequestration and precipitation. The estimation coefficients of annual mean temperature and slope were all positive (+) and significant at a 1% level. On one hand, higher temperature and more precipitation would result in a higher survival rate of afforestation and extension of vegetation boundaries [61]. On the other hand, rising temperatures will increase species diversity [62] and alter vegetation types, e.g., converting coniferous forests to broadleaved forests [63], increasing carbon density and forest carbon sequestration in the region. Additionally, the model showed that the greater the slope, the less the human activity, which could contribute more to forest growth and consequently increased forest carbon sequestration. The estimation coefficients of aspect and land urbanization rate did not pass the significance *t*-testing. That means aspect and land urbanization rate had no significant effects on forest carbon sequestration in Yunnan. The regression result of per capita GDP growth rate was not significant; thus, the change in carbon sequestration was unrelated to the level of economic development. The regression results of land urbanization rate and per capita GDP indicate that the level of economic development was not the main reason for the regional differences in the impact of GFG government expenditure on forest carbon sequestration.

## 4. Discussion

Notably, the regression results of Model (3) show that the estimation coefficients of Yangbi Yi Autonomous County, Yongping, and Huaping County are negative (−) and pass the significance *t*-test. These counties (cities or districts) are mainly distributed in the east of Lijiang City and the southwest of Dali Bai Autonomous Prefecture. This study draws a scatter plot of GFG government expenditure and forest carbon sequestration in the three counties (see Figure 5). The plot shows that the forest carbon sequestration first increased then decreased with increasing GFG government expenditure, which defies common sense. Generally, GFG government expenditure in the three counties that passed the significance t-test with negative (−) sign of the coefficients and the 51 counties (cities or districts) that did not pass the significance t-test did not have a promoted growth of the forest carbon sequestration. This may be because these 54 counties (cities or districts) did not effectively improve the local ecological environment due to inefficient use of the funds.

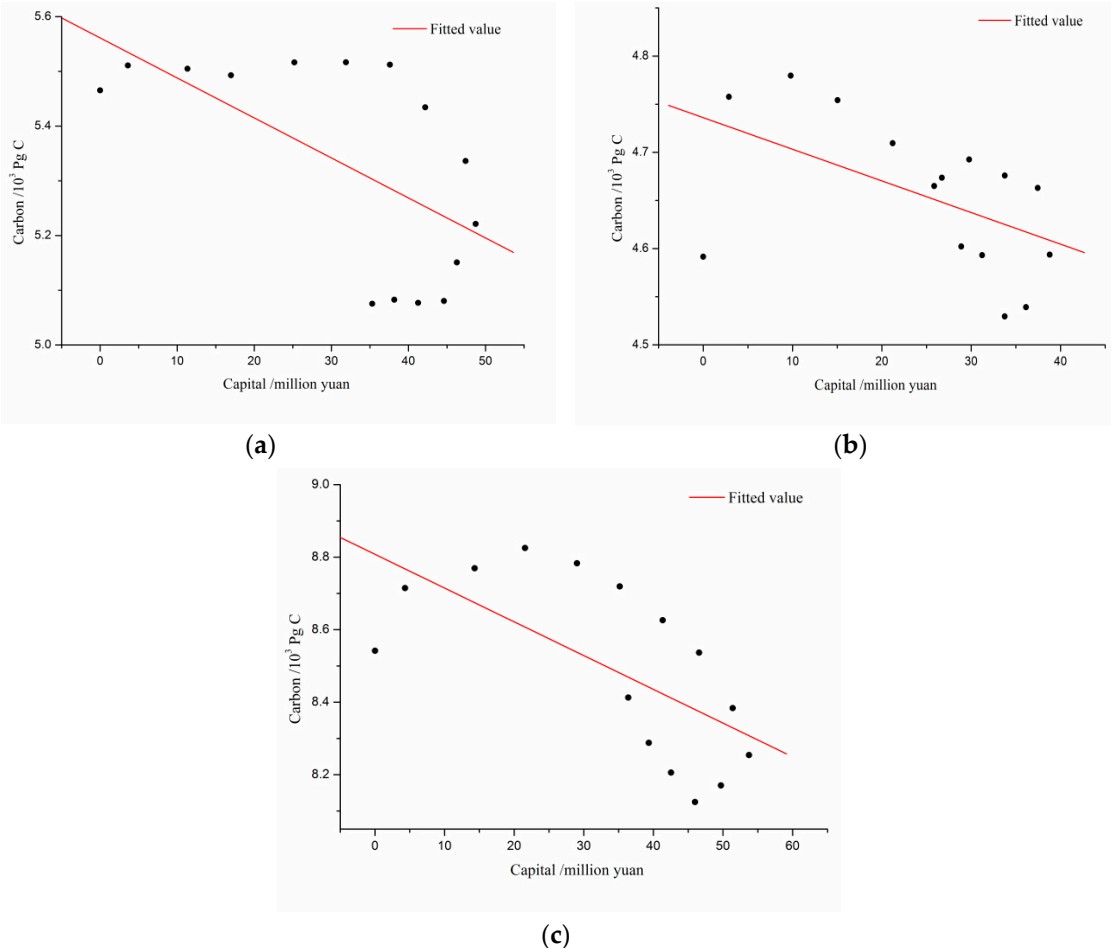

**Figure 5.** Scatter plot of forest carbon sequestration and government expenditure on the GFG. (**a**) Huaping County; (**b**) Yangbi Yi Autonomous County; (**c**) Yongping County.

A deficiency of this study is that some uncertainty remains in the estimation results of forest carbon storage in Yunnan Province. This is manifested as: (1) the carbon density of Shangri-La County was only used to evaluate the vegetation cover types in the entire province; the spatiotemporal heterogeneity of carbon density of the same vegetation cover type was not considered. (2) It is assumed that the soil carbon density of each vegetation cover type will remain unchanged for many years over the study period, increasing the uncertainty of soil carbon storage estimation results. Some studies have shown that even with unchanged vegetation cover type, the soil organic carbon content will change with time depending on the vegetation growth status, thus affecting soil carbon density [64]. Therefore, to more accurately evaluate the forest carbon storage in Yunnan Province, the local carbon density must be corrected by combining appropriate variables in future research, or more accurate dynamic carbon density can be obtained through sample plot measurement. Further, due to certain limitations of the samples and methods, this study does not deeply explore the function and influence mechanisms of the GFG government expenditure on forest carbon sequestration. Additionally, the impact of GFG government expenditure is multifaceted, involving soil and water conservation, wind and sand prevention, and the livelihoods of farmers; these provide another direction for further exploration.

## 5. Conclusions

The GFG has been operating for over 20 years. Under the background of global promotion of increasing carbon sequestration and reducing emission, evaluating the effectiveness of government expenditure on the GFG for providing forest carbon sequestration is necessary. Using the panel data from 2001–2015 of 129 counties (cities or districts) in Yunnan, this study estimated the impacts of GFG

government expenditure on forest carbon sequestration and explored the regional differences in the impact of GFG government expenditure on forest carbon sequestration. The conclusions are as follows:

(1) In this study, the least squares dummy variables method (LSDV) was used to estimate the effectiveness of GFG fiscal expenditure in Yunnan. The marginal contribution of the GFG government expenditure on forest carbon sequestration was uncovered effectively. The results indicated that GFG government expenditure effectively increased the supply of forest carbon sequestration and improved the regional ecological environment. This study method is expected to provide reference for evaluating the effectiveness of ecological project in relevant areas.

(2) Although the GFG policy has made remarkable effects in providing forest carbon sequestration at the provincial level, the policy effects at the county level remain to be tested. This study found that there were significant regional differences in the increase of forest carbon sequestration in the government expenditure of the GFG, which was mainly due to natural environments, especially initial forest resource endowments.

(3) This study maintains that, due to differences in initial forest resource endowments, to achieve a sustainable and significant increase in forest carbon sequestration, the government should increase the investment in the 75 counties (cities or districts) that had positive (+) regression coefficients and passed the significance *t*-test. Those counties (cities or districts) include Diqing Tibetan Autonomous Prefecture in the northwest, Baoshan City in the west, Honghe Hani and Yi Autonomous Prefecture in the south, and Wenshan Zhuang and Miao Autonomous Prefecture in the east. For those counties (cities or districts) that did not pass the significance *t*-test, the use efficiency of the government expenditure on the GFG should be improved to effectively increase forest carbon sequestration and improve the ecological environment.

**Author Contributions:** Conceptualization, Y.L., Z.D. and S.Y.; methodology, Y.L., Z.D., Y.D.; software, Y.L.; validation, S.Y. and M.H.; Formal analysis, Y.L.; investigation, S.Y. and M.H.; resources, S.Y.; data curation, Y.L. and Y.D.; writing—original draft, Y.L.; writing—review & editing, Z.D., S.Y. and M.H. visualization, Y.L.; supervision, S.Y.; project administration, S.Y. All authors have read and agreed to the published version of the manuscript.

**Funding:** This research was funded by Monitoring and Evaluation of the Grain for Green project and Its Technical Optimization, grant number 201504424.

**Acknowledgments:** We would like to give our appreciation to the Returning Farmland to Forest Office of Yunnan Province for its data support.

**Conflicts of Interest:** The authors declare no conflict of interest.

## Appendix A

The regression results of Model (3) are shown in the support information.

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
