# Peer review of "Did Government Expenditure on the Grain for Green Project Help the Forest Carbon Sequestration Increase in Yunnan, China?"

_land, doi:10.3390/land9020054_

Round 1

Reviewer 1 Report

This study addresses an interesting question: how can influence fiscal expenditures in environmental programs into forest carbon sequestration at a regional level. The authors have used a panel data model in order to test several questions regarding the relations between both issues in a Chinese province. The aims and objectives of the paper fit in the editorial policy of this journal. Although I think the authors have made a worthy effort, my overall recommendation is that the manuscript should not be accepted now in its present form. My comments are listed as follows:

General comments

Methodology: although the authors provided the reasons to choose the methods used in this manuscript, it is a quite strange than no bibliographic reference about LSDV appears in the Methods nor the Discussion Section. While the capital stocks measures are measured annually, it is not clear if carbon sequestration has been computed in each year. The authors must clarify this issue, and if the carbon is measured annually, the authors should explain which method has been used to compute it (see https://doi.org/10.1080/17583004.2017.1306407) Econometric models have been not well explained (see main comments) The manuscript is focused on the case study. The readers would expect that a conclusion or consequence of this research could be applied in other countries or in other situations.  authors can also refer to those papers:

10.1007/s10342-014-0828-0   10.1016/j.envsci.2011.11.001   10.1016/j.foreco.2014.09.038

Main comments

88 Why “potential” carbon sequestration contribution? Figure 1: the authors must explain the variables and the colours used in this Figure. 146: in other Sections (i.e., l. 221), the authors hypothesized about the sign of the components of statistical models. I think they must do the same with this equation. What is the expected sign about “ln carbon” and “ln capital”? In this eq., is “Llncarbonit” or “lncarbonit”? 153: the acronyms have been not defined 283-284: the consequences of this positive relation are not clear. Maybe the authors could explain in this issue in detail 289-291: dummy variables cannot be introduced in the Results Section. They should be described in the Methods Section, and the interaction terms must be defined appropriately. The word “efficiency” appears profusely in the Results Section, but it has not previously described. What means “positive efficiencies” (l. 302-303). And “inefficient” counties? (l. 306) Table 3. Models (1), (2), and (3) should be defined before (and their equations). Besides, interaction terms do not appear; why? Finally, many have not to be defined previously. In short, a reader cannot replicate this methodology because the explanations are scarce and dispersed along the manuscript 314: if deforestation is not allowed, how forest carbon sequestration can decrease along the time? The authors do not include a Discussion Section. No comparison with other studies is provided. This Section must be re-written.

Reviewer 2 Report

Very interesting and substantial but serious improvements are recommended. Many points/ passages should be restructured, clarified or deleted.

See comments attached.

Reviewer 3 Report

The structure of the introduction is odd. Typically, the introduction is used to provide an overview of the topic, outline its importance, indicate why the research is important, and list the objectives of the work. Some discussion is part of results. I would recommend separating discussion and conclusion sections. In the conclusion section include limitations and chart out an agenda for future research. New and clear structure is needed.

Your literature review is good but it excludes some literature about effectivity of subsidies in the theoretical background and in the discussion. It would be critical to include that literature as part of your literature review. For example the approach in Juutinen, A., Ahtikoski, A., Lehtonen, M., Mäkipää, R., & Ollikainen, M. (2018). The impact of a short-term carbon payment scheme on forest management. Forest Policy and Economics, 90, 115-127.

Reviewer 4 Report

I read with great interest the manuscript “Did fiscal expenditure on the Sloping Land  Conversion Program help increase the forest carbon sequestration in Yunnan,China?.” which examines the issue of government spending on  the individual contribution of a public good . In general, the manuscript does not live up to its expectations. The manuscript needs proper proof reading, a literature review is needed, a better justification of the modeling choice will be welcome while a rather less wordy writing style will benefit the manuscript.

Since the SLCP is designed to avoid soil erosion, isn’t necessary to define the benefits of the scheme according to its prime objective, namely avoid soil erosion. Then you may add side-effects as the carbon sequestration.

My reservation are :

Title: Government spending (or expenditures) is preferred term instead of fiscal expenditures. Title: Why the general reader should be familiar with what the “Sloping land Conversion Program” is? Abstract: lines 14-15 vague. Abstract: carbon sequestration from cities? Page 2: lines 46-47 vague Page 2: what is the meaning of “invested in special fiscal transfer payment funds”? Page 2: why the authors use a double referencing system? (both author-date and numbered) Line 58: what is GWR? All the abbreviations need definition Lines 70-71: incomprehensive Line 108: What is the difference between ecological and environmental? Line 113: What is “returned farmland”? Lines 126-127: incomprehensive Line 143: why equation (1) uses such a messy way to put down lag variables and not use as in (3) t-1? Line 143: the control variables in (1) are not defined Line 171: what are the (IGBP, UMD, LAI, BGC, and PFT? Lines 361-362: incomprehensive Line 369: what is : regional ecological  environment? Lines 398-404: Figures (analysis) in the conclusions?

Round 2

Reviewer 1 Report

Unfortunately, after reading the authors’ responses, this manuscript should not be accepted. My comments are listed as follows:

General comment:
• Under my view, the authors have not adequately responded to the majority of the questions raised in the first revision

Main comments
• It is not clear if carbon sequestration has been computed in each year. Regarding the INVEST model, it is not acceptable considering the same density for the different carbon pools considered. Besides, nothing is said about the different carbon content of each species.
• The methods are not at all precise, and the bibliographic references provided are not sufficient. A reader cannot understand why these econometric methods have been chosen. In this new version, some elements belong to eq (3) have been defined at the county level, but not by year. How have been some variables calculated annually (GDP, land urbanization rate)?
• I'm afraid I have to disagree with some explanations provided. For example, the use of “efficiency” is not suited. “Significative” would be better
• The statements regarding the decrease of the carbon along time show an evident weakness of the model.
• The Discussion Section is not acceptable: there is no comparison between the results obtained and other literature related.

Reviewer 2 Report

Everything is ok now

Reviewer 4 Report

The authors have addressed all my reservations. No more comments